# The Use of Fiber-Reinforced Polymers in Wildlife Crossing Infrastructure

**Matthew Bell [1], Damon Fick [2],\* , Rob Ament [1] and Nina-Marie Lister [3]**

[1] Western Transportation Institute, Montana State University, Bozeman, MT 59717, USA; matthew.bell8@montana.edu (M.B.); rament@montana.edu (R.A.)

[2] Norm Asbjornson College of Engineering, Montana State University, Bozeman, MT 59717, USA

[3] School of Urban and Regional Planning, Ryerson University, Toronto, ON M5B 2K3, Canada; nm.lister@ryerson.ca

\* Correspondence: damon.fick@montana.edu

**Abstract:** The proven effectiveness of highway crossing infrastructure to mitigate wildlife-vehicle collisions with large animals has made it a preferred method for increasing motorist and animal safety along road networks around the world. The crossing structures also provide safe passage for small- and medium-sized wildlife. Current methods to build these structures use concrete and steel, which often result in high costs due to the long duration of construction and the heavy machinery required to assemble the materials. Recently, engineers and architects are finding new applications of fiber-reinforced polymer (FRP) composites, due to their high strength-to-weight ratio and low life-cycle costs. This material is better suited to withstand environmental elements and the static and dynamic loads required of wildlife infrastructure. Although carbon and glass fibers along with new synthetic resins are most commonly used, current research suggests an increasing incorporation and use of bio-based and recycled materials. Since FRP bridges are corrosion resistant and hold their structural properties over time, owners of the bridge can benefit by reducing costly and time-consuming maintenance over its lifetime. Adapting FRP bridges for use as wildlife crossing structures can contribute to the long-term goals of improving motorist and passenger safety, conserving wildlife and increasing cost efficiency, while at the same time reducing plastics in landfills.

**Keywords:** fiber-reinforced polymer; bridges; wildlife-vehicle collisions; composites; wildlife crossing; green infrastructure

## 1. Introduction

For several decades, ecologists and engineers have been exploring new methods and adapting existing techniques to more effectively address mitigation measures that address motorists' safety and large species conservation as a result of wildlife-vehicle collisions (WVCs). There are over one million WVCs with large animals every year in the U.S. that result in substantial property damage, personal injuries, and fatalities [1,2]. There are currently many techniques for mitigating WVCs, including wildlife crossing infrastructure (i.e., underpasses and overpasses), wildlife signage, reflector posts, fencing, and animal detection-driver warning systems. Some are highly effective, while others are not [3,4]. When mitigation structures are properly located and designed, they can reduce WVCs from 80% to 99% [3–7]. Because of the proven effectiveness of crossing structures with fencing to reduce wildlife mortality, increase motorist safety, and maintain connectivity across transportation systems for all sizes of animals, they are often the preferred mitigation measure around the world.

Overpasses and other large crossing structures are the desired infrastructure to meet certain wildlife and highway project design considerations and have proven essential to maintain demographic

connectivity for particular species [8]. Although usually more expensive than underpasses [3], overpasses are also frequently chosen by some species [9].

The length and the width of overpasses continue to challenge engineers, architects and ecologists. Some overpasses are required to span six or more lanes, including Canada Highway 1 in Yoho National Park and Interstate Highway 90 in the Cascade Mountains of Washington. Some newer designs are anticipated to exceed lengths spanning 10–12 lanes. The proposed wildlife overpass on Highway 101 in Liberty Canyon, California, will require bridge spans up to 60 meters (m) and be the largest wildlife overpass ever built, as seen in Figure 1 [10]. Common widths of overpasses have been designed from 30 to 60 m and even wider. These geometry requirements can result in massive and relatively uneconomical structures. Many existing wildlife overpass structures are constructed to support heavy loads that incorporate excessive backfill to host native habitats, such as forests. This design feature adds substantial weight to the static and environmental loads the structure is required to support. Supporting these loads over multi-lane roadways further results in relatively high costs compared to underpasses or other less-effective mitigation measures. Because of the costs, the siting of these structures is especially challenging as they can only be provided sparingly across a large area where WVCs commonly create safety issues for drivers. Recent price tags for wildlife overpasses near Banff, Alberta, Canada, cost over $4 USD million, and current estimates for the Highway 101 overpass in Liberty Canyon are estimated at over $50 USD million [3,11].

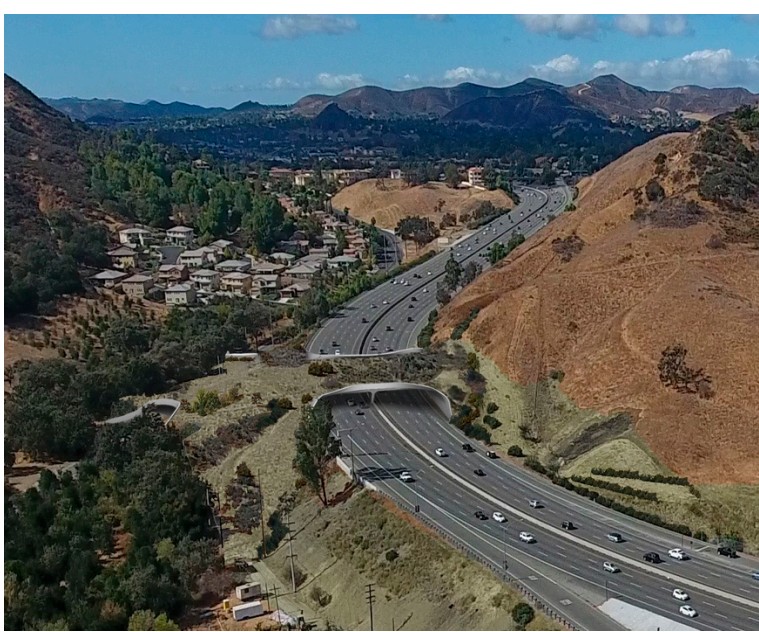

**Figure 1.** Rendering of a proposed Liberty Canyon wildlife crossing structure over Interstate 101, Los Angeles County, California. The proposed crossing will span 10 lanes of traffic. Photo credit: Resource Conservation District, Santa Monica Mountains, Clark Stevens Architect, Raymond Garcia Illustrator.

Not all crossing structures are located in forested environments or are designed for a focal species requiring hiding cover and other types of dense or tall vegetation. Nor will most mitigation locations be on highways with four or more lanes. In fact, almost 90% of all WVCs in the United States occur on two lane roads [1]. For example, in the largely rural state of Montana, nine out of the top ten WVC hotspots during the fall migrations of wildlife occur on two-lane highways [12]. Thus, many, if not most, overpasses and crossing structures will address short spans where larger focal species will only briefly spend time using the crossing.

Most, if not all, North American overpass structures have been designed by engineers using concrete, either precast or cast in place, steel arches, or some mix of steel and concrete. The landscape

surfaces are often designed and commissioned later, although there is evidence that the success of these projects may require more integrative design approaches in which the landscape components are considered together with the wildlife overpass [13]. These existing methods have limitations, mainly the duration of construction, which results in traffic control and detours for up to six months. Heavy equipment increases costs due to the size and weight the cranes are required to lift. The unique placement and surroundings of each structure requires the overpass to be designed independently which reduces the efficiency of these structures.

In addition to the restrictions with construction and design, concrete and steel are less durable due to environmental freeze-thaw cycles that result in cracking, salt intrusion, and reinforcement corrosion. For steel structures, regular inspections are required to identify potential fatigue cracks and/or corrosion. For steel members made from non-weathering steels, routine painting is required maintenance. At the end of their service life, these permanent structures often require significant rehabilitation and extended maintenance, making bridge replacement a more economical option for bridge owners. Moreover, approximately 5% of global $CO_2$ emissions originate from the manufacturing of cement, and it is the third largest source of carbon emission in the United States [14].

Published research on bridge designs and materials for wildlife crossings is limited and suggests relatively little innovation has occurred over several decades [13]. Because overpasses are incorporated into highway mitigation designs for species-specific preferences, increased motorist safety, and to support genetic exchange; the need for new, resourceful, and innovative techniques is warranted. This paper explores the promising application of fiber-reinforced polymers (FRPs) to wildlife crossing structures that can potentially provide a less expensive and more adaptable approach to mitigating WVCs. If FRP crossing structures prove successful, transportation planners, engineers and landscape architects will be able to design and implement more wildlife crossings using an integrated planning and design approach to provide safe passage for motorists and wildlife, alike [13].

This article (1) introduces the benefits and limitations of FRP materials in crossing designs and how they can be applied to many different types of wildlife crossing infrastructure; (2) identifies design-based opportunities and procedures for incorporating recycled plastic material into wildlife crossing infrastructure designs; and, (3) evaluates and articulates the political and administrative processes that will help facilitate the adoption of plastic bridge designs by federal, state and local transportation agencies in North America.

## 2. Fiber-Reinforced Polymers

Fiber-reinforced polymers are a composite material of structural fibers set in a mold of thermoset resin (Figure 2). These resins do not get soft at elevated temperatures and so they restrain the fibers against buckling to allow the transfer of shear stress between fibers [15,16]. Polyesters, vinyl esters, and epoxies are the most commonly used thermosetting plastics for FRPs, but also include other synthetic, bio-based, or recycled polymers to adhere fibers together [17,18]. The type of resin selected depends on the purpose of the structure. Each resin has different chemical properties and strengths. Some materials are more resistant to environmental elements and can increase the life expectancy of a structure. The type of resin directly relates to the beneficial properties of FRP structures to resist various physical (e.g., wheel rolling, collisions, debris) and environmental (e.g., moisture, oxidation, ultra-violet [UV] rays) impacts [16].

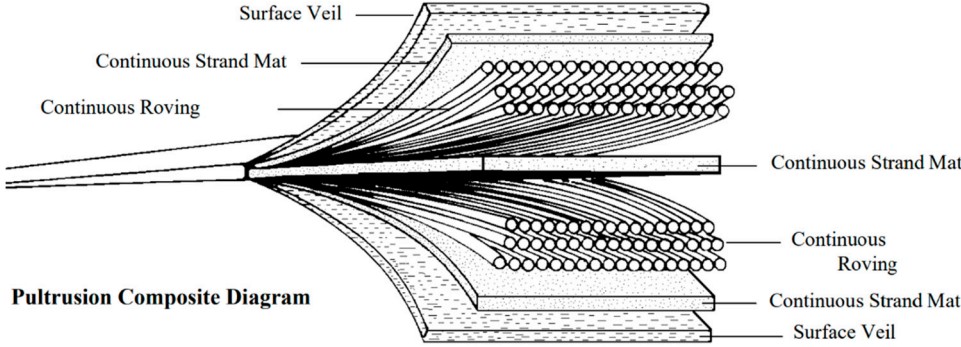

**Figure 2.** General configuration of structural fibers distributed throughout the thermoset resin. Figure credit Creative Pultrusions, Inc.

Most of the strength of an FRP comes from the choice of fibers used within the mold. Glass is the most commonly used fiber, but carbon and aramid fibers are superior although they generally cost more than glass. Fibers are confined to locations that mimic thick metal strands, randomly assorted within the mold, or layered down as fiber mats matrixed within the resin. This application of fibers can be compared to rebar in reinforced concrete, but at a smaller scale, and dispersed throughout the entire mold. The material and configuration of fibers is based on the desired structural strength requirements.

A common concern with polymer composites is their time-dependent mechanical properties. It is well known that laminate layup, stress level, frequency of loading, humidity, and temperature can affect the FRP behavior over time [19]. Due to the lack of long-term experimental data for different FRP materials, deformations are often estimated based on results of short-term, accelerated testing conditions [20]. Theories for characterizing the mechanical properties specifically for civil engineering applications are well documented [21]. These and other advancements in the design of FRP composites make them ideal for applications in civil infrastructure.

### 2.1. Benefits of Fiber-Reinforced Polymers

Fiber-reinforced polymers can outperform concrete and steel because of their dimensional stability, high strength and light weight properties. Case studies show the average FRP bridge is half the weight of a steel bridge with the same strength and is five times lighter than its concrete equivalent [22–24]. Additional benefits of a lighter structure are reductions in energy and construction costs (i.e., transportation, erection, supporting foundations, construction time). Another quality of FRP structures is that they are water and corrosion resistant, making them suitable for use in marine settings (e.g., lock gates, pilings, decking). Resins can absorb water through osmosis at a microscopic level, but the process is reversed when the FRP is dried [16]. Applications of moisture resistant resins can also be applied to the outside of the FRP structure if the use of these resins become cost prohibitive for the entire mold.

Depending on the properties of the resins and fibers used within FRP structures, they can be fire and ultra-violet (UV) resistant, electromagnetically transparent, impact resistant, have low thermal conductivity, provide no electrical conductivity, and have low maintenance costs [15,16,23]. They are highly durable, have a life span up to 100 years, and require little to no maintenance for their entire life cycle; allowing FRP structures to pay for themselves over time [24–28]. At the end of the bridge's service life, several technologies for recycling and applications for reuse are available [29].

### 2.2. Production of Fiber-Reinforced Polymer Products

There are two strategies for making FRP materials for infrastructure construction. The first method creates structural cross-sections through the process of pultrusion; where the fibers and resin are pulled through a mold simultaneously to create continuous beams (Figure 3). The fibers pulled through a pultrusion machine can either be individual strands, or multiple strands woven together to form matrix

mats. The fibers are pulled into a heated area where they are combined with the resin. Afterwards, the resin and fibers are pulled through a mold to make the desired geometric cross-section. They can be formed into bars, plates, structural tubing, channels, and any type of custom girder shapes. After the FRP member is formed into its desired shape and cooled, it can be cut to any length. Forming the structural members is an intensive process but is extremely efficient when large quantities of a cross-section are needed. Commonly made of recycled polymers, these methods are being adopted as a solution for replacing old and deteriorating structures [30]. Pultrusion molds are assembled using steel and lumber construction methods, commonly connected with stainless-steel bolts. An example of an FRP pultrusion-style pedestrian bridge can be seen in Figure 4.

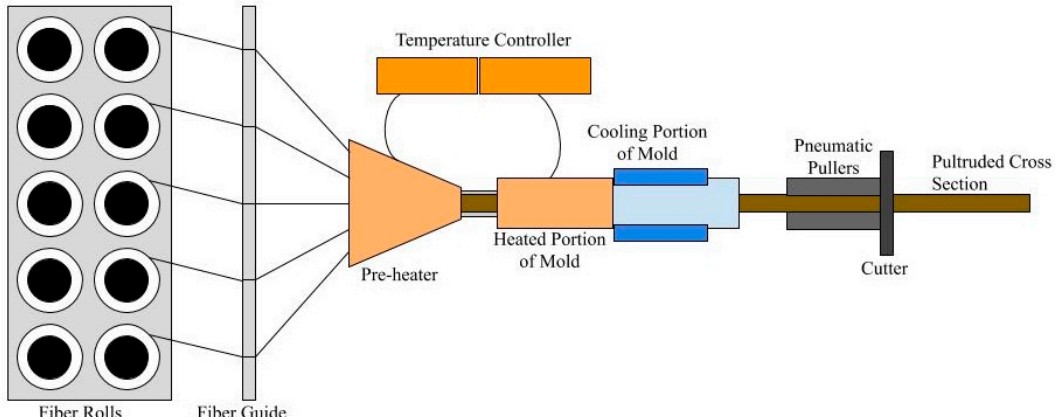

**Figure 3.** Schema of how pultrusion members are formed. Fibers are pulled through the guide device and into the pre-heater where they are combined with the resin. The resin and fibers are then pulled through the mold of the desired cross-section design. After cooling, the pultruded member is cut to length. Figure credit Matthew Bell.

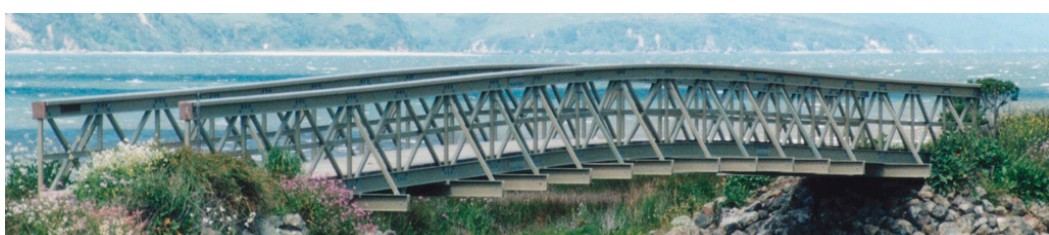

**Figure 4.** Example of pultrusion-style pedestrian bridge in Marshall, CA. The bridge spans 29 m and is 1.8 m in width. With a live-load design of 2.83 kilopascals (kPa), or 60 pounds per square-foot (psf), the FRP members are connected with galvanized steel bolts. Photo credit: Creative Pultrusions, Inc.

The second fabrication method is the use of vacuum assisted resin transfer molding; a process that pumps resin through custom shaped molds with the incorporated fiber layouts (Figure 5). Manufacturers create a mold so all the fibers can be laid out within the structure. The entire mold is sealed air-tight and vacuum pumps are attached to one end, while intake valves submerged in resin are attached to the other. When the vacuums are activated, they extract the air out of the mold and pull the resin through. This process is extremely efficient at impregnating all the fibers with resin by removing the air bubbles within the mold. The molds can result in free-formed standalone FRP bridge spans (uni-molds) installed on concrete and/or steel supports (Figures 6 and 7), or hybrid structures that incorporate steel and concrete materials to create the bridge span. These molded bridges are able to span long distances and support large loads while removing fasteners throughout the structure, which removes weak links in the bridges. This process enables the possibility of complex cross-section designs and the integration of other materials. Core inserts can be applied in geometric formations

(e.g., squares and hexagons) to increase structural efficiency. For these cases, the fibers are arranged around the core material to produce strong, lightweight, and durable FRP structures.

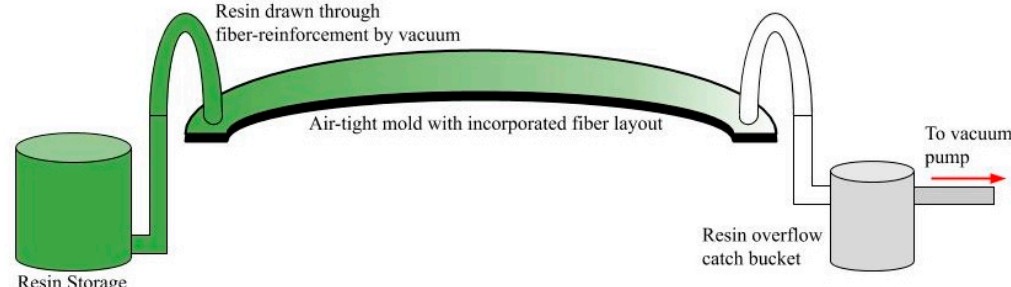

**Figure 5.** Schema for how vacuum assisted resin transfer molded structures are formed. A vacuum pump removes all the air from the sealed mold and pulls the resin through associated fiber layout. Once the mold is completely impregnated with resin, the vacuum pump is removed and the mold is sealed off, allowing the FRP structure to cure. Figure credit Matthew Bell.

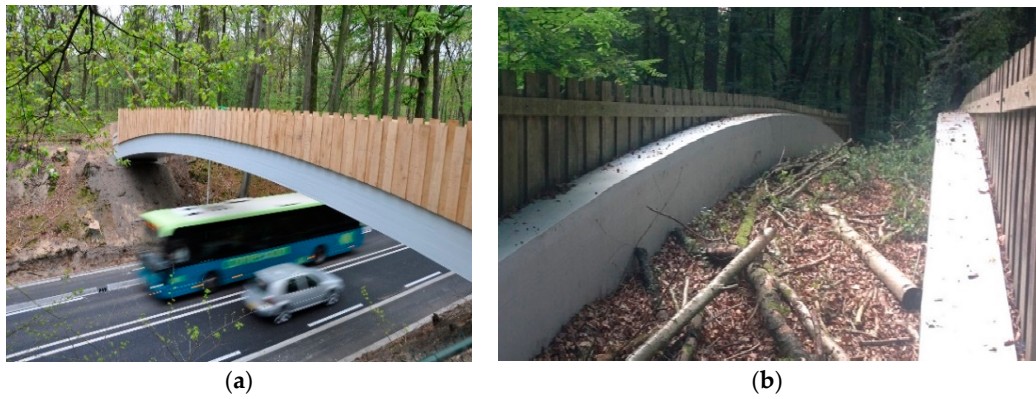

(**a**)      (**b**)

**Figure 6.** Example of vacuum assisted resin transfer FRP bridge in Rhenen, The Netherlands. This overpass on the N-225 highway is the first FRP structure designed for wildlife use. (**a**) The bridge dimensions are 24 m × 2 m. Photo credit: FiberCore Europe. (**b**) The top of the structure provides a wildlife-friendly surface. It is covered with topsoil and biomass, including plant material used from the surrounding area. Photo credit: Matthew Bell.

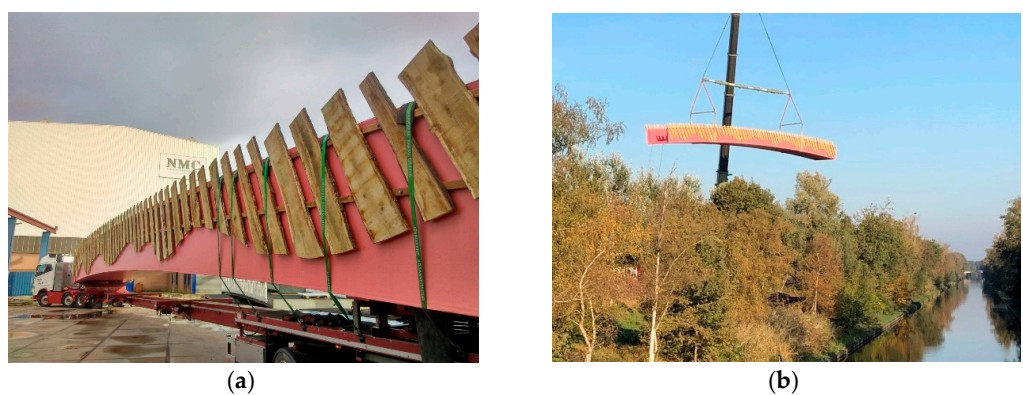

(**a**)      (**b**)

**Figure 7.** A uni-mold wildlife overpass with dimension of 36 m × 3.5 m in Eindhoven, The Netherlands. (**a**) The bridge was shipped on the back of a truck in one piece after manufacturing. (**b**) The single bridge span weighs 29 tons and was installed within a few hours using a single crane. Photo credits FiberCore Europe.

FRPs support modular construction design, variation in the way the fibers are laid out, and different methods of fabrication [18,24,27,30]. The dimensional constraints of FRP products are limited

by transportation logistics, not in the structural properties and technology itself. In principle, there is no limit to the dimensions of the FPR elements in a bridge design. The maximum capabilities of this innovative material are not fully understood and require additional research [22].

## 3. Using FRPs for Wildlife Crossing Infrastructure

The majority of current FRP infrastructure follow the principles of accelerated bridge construction (ABC). This approach minimizes traffic disruptions during bridge construction, promotes traffic and worker safety, and improves the overall quality and durability of bridges [31]. The structural members are light and include simple connections for easy erection. Conventional methods for building wildlife crossings around the globe are centered primarily on concrete and steel structures. Although FRP wildlife infrastructure may adopt some of these techniques, they are streamlined by lighter materials and increased durability due to the polymer molding process. Unlike concrete and steel designs, uni-mold FRP wildlife bridges can be shipped by the manufacturer to the construction site as a single piece and lifted into the final bridge configuration using standard shipping and construction equipment (Figure 7).

Depending on site-specific landscape attributes and wildlife requirements, FRP wildlife bridges can be designed to support living plant material—a natural vegetated surface—on the deck that is attractive to, and suitable for, the target species. Structures can be designed to support the loads of topsoil or soil-less surfaces which include: aggregate fill; geotextiles; a variety of vegetation (e.g., diverse selection of trees, shrubs, forbs and grasses); rain, snow and/or ice accumulations; sound/light barriers; research equipment, etc. The surface environment for a crossing structure should be attractive to wildlife, mimic adjacent habitat (in terms of vegetation and substrate matter), and provide a medium for locally-adapted vegetation establishment and growth [32,33]. Because there are a variety of environments where WVC mitigation is needed, structural typologies for FRP wildlife crossings will require creative and innovative designs. Bridge geometries and assembly configurations that are adaptable will help ensure their designs are ideally suited to their habitats.

Although wildlife overpass structures have been designed to support deep soils (e.g., one meter) and large plant communities (e.g., forests), it is not always required. For example, in alpine, sub-alpine, grasslands or desert landscapes, a variety of substrates and plant materials can be deployed as appropriate to the context. With the lighter FRP material, it is also possible to integrate green-roof technology using lighter-weight substrates to provide functional habitats equipped with waterproof membranes and proper drainage. This may produce significantly smaller masses the structure is required to support. Designing slender and lighter FRP structures will require greater attention to dynamic excitations caused by wind or live loading on the bridge. Due to the high-strength FRP materials, structures are often controlled by these stiffness characteristics. An FRP wildlife overpass structure has the potential to have a much lower structural profile than traditional materials and can reduce the amount of material required to build the FRP bridge spans—making them more readily designed to fit into the local landscape.

### 3.1. Adaptive FRP Wildlife Crossing Infrastructure Alternatives

### 3.1.1. Modular Uni-Mold Bridge Design

A modular arch bridge design that consists of three molds, two different end pieces and a middle section, is shown in Figure 8a. This prototype is capable of spanning a three-lane road (~15 m) and can be constructed together to make adaptable wildlife overpasses. This design is applicable from two-lane to multi-lane highways with medians, where intermediate supports could be constructed. The middle section shown in Figure 8a can be duplicated in the transverse direction to create crossings of variable widths as shown in Figure 8b. The FRP modular crossing can be designed with a large radius arch that will provide adequate traffic clearance and connect at the road level, which will significantly decrease

costs in design, materials, and transportation. The shallow arch design allows ample visibility for animals and drains water away from the supported bridge.

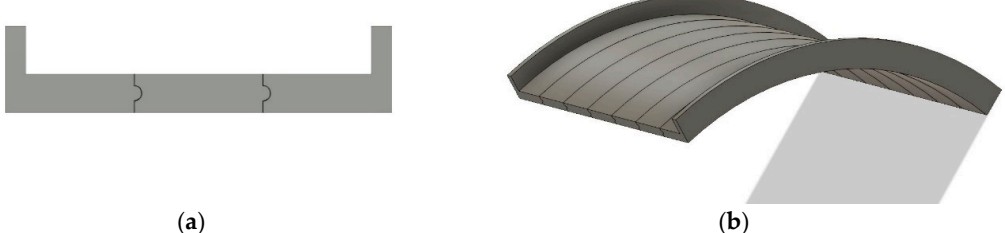

(**a**)　　　　　　　　　　　　　　　　　　　　　　　　(**b**)

**Figure 8.** Example of a uni-mold superstructure design to make a modular and adaptable wildlife overpass. This method uses three different FRP structures that can be combined to make versatile wildlife overpasses. (**a**) Cross-section of three FRP mold proto-types and how they can be connected together to build adaptable bridges. (**b**) A rendering that shows how adding additional middle sections allows the bridge to be made to any desired width. Figure credit Matthew Bell.

These modular units will need to be installed onto separately constructed bridge abutments to reduce FRP crossing length and to meet highway setback and vehicular clearance requirements. The foundations can be built using current techniques and materials, or new designs that combine traditional materials with FRPs that support the overpass. A narrow individual unit width will allow molds to stack and ship on North American highway systems. In order to reduce traffic disturbance, the FRP bridge abutments and foundations are often constructed without traffic interruptions or detours. The modular units can then be transported and installed rapidly, reducing traffic disruptions. Free-formed FRP overpasses are able to adapt within a changing landscape. If there are environmental factors that influence changes in wildlife migrations, movement routes or hotspots of WVCs, the bridge is able to be deconstructed and moved to a new location—modularity and adaptability are key design elements noted in the need for materials innovation [13,34].

### 3.1.2. Hybrid Composite FRP Beams

Concrete-filled FRP tubes (CFFTs), and other types of FRP hybrid composite beams (HCBs), are light-weight empty FRP exoskeletons that are positioned without the use of heavy equipment, and then filled with concrete after they are installed. The outer FRP structure protects the concrete from the environmental elements and therefore increases the service life of the structure. This process reduces the initial weight of the beams and allows multiple beams to be shipped on one truck. The footings and abutments for the beams can be cast-in-place concrete and connected together with FRP paneling. The beams and the panels are the only structural components required (Figure 9). Many current applications of HCBs exist that support the static and dynamic loads of traffic flow and span over 30 m. These FRP beams are becoming more popular because of their quick installation time, high strength, light weight and long life-cycle when compared to traditional concrete and steel beams. The McGee Bridge (8.5 m × 7.6 m) replacement project in Anson, Maine, was completed in 12 working days using CFFTs; this included the removal of the old bridge being replaced. Commercial champions of this technique claim a CFFT bridge span can be completed in as little as three days [35].

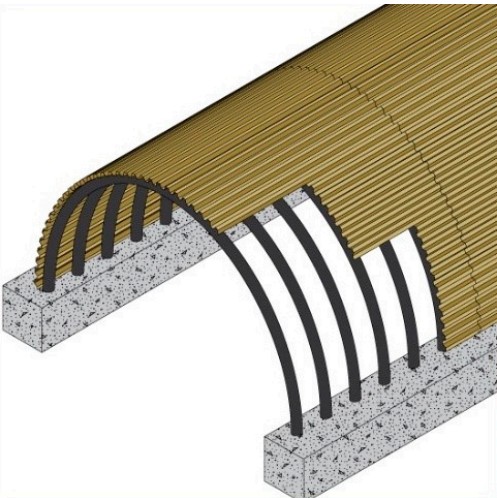

**Figure 9.** General design for a concrete-filled FRP tube (CFFT) bridge with a single-radius arch design. The FRP tubes are cast in place, connected with FRP panels, and then filled with concrete. Figure credit Matthew Bell.

These cast-in-place CFFT arches are adaptable to all road types. Consisting of single or double radius arch designs, bridges can be built to span all lanes of traffic or use the median as a support location to connect two smaller arches. Although larger FRP tubes that span over 60 m are being designed and tested off-site, tubes for shorter spans can be constructed on location, reducing the costs of expensive transportation logistics. These structures have been designed to satisfy the American Association of State Highway and Transportation Officials (AASHTO) requirements for traffic loading and are promising structures that can be adapted as wildlife over- and underpasses [36].

### 3.1.3. FRP Culverts

Another structure used for wildlife underpasses is steel and concrete culverts, ranging from 0.3 m to over 3 m in height. These culverts are not a cost-efficient alternative to overpass structures, but rather are preferred by certain animal species (e.g., reptiles and amphibians). Applications of small FRP culverts are currently used to move water and other materials, and are becoming more common because of their resistance to chemicals [37,38]. Generally, concrete and steel wildlife culverts can be replaced with more durable and longer lasting FRP moldings. They are a promising alternative to traditional steel culvert design and require less maintenance. The FRP has better mechanical performance than a steel culvert in terms of load-bearing capacity and fatigue issues [28]. The FRP culverts is comparable to steel's ability to span long distances with shallow soil cover. Along with structurally outperforming steel culverts in many aspects, they are also economically feasible and provide more durable and sustainable performance during their service life [38].

### 3.1.4. Recycled Polymer Applications

Recycled plastics lack the structural properties that are found within new synthetic resins, but have the same benefits of withstanding environmental elements and resisting degradation [17]. This creates challenges for using them in FRP structures because they cannot be molded in the same way as virgin resins. Therefore, recycled plastics are commonly used in non-structural applications because it is more difficult to include complex fiber distribution throughout the mold. An example of a non-structural application is wildlife exclusionary fencing. These fences funnel animals to under- and overpass crossing structures and are key to their effectiveness in reducing WVCs [39]. Wildlife jump-outs or escape ramps are placed along the exclusionary wildlife fencing so animals can jump to safety if they end up between the fences lining the roadway (Figure 10). These escape ramps allow animals to walk to the top and jump down to safety but are high enough so animals are deterred from jumping up onto

the roadway side of the fence. Another type of fence in overpass crossings is the barrier fence that is designed to line each side of an overpass structure to keep wildlife safely on the overpass, dampen noise on the overpass from traffic below and obstruct/block light and glare from passing vehicles, particularly at night. Vehicle noise and headlights are known to deter wildlife from using the overpass structures [33,40].

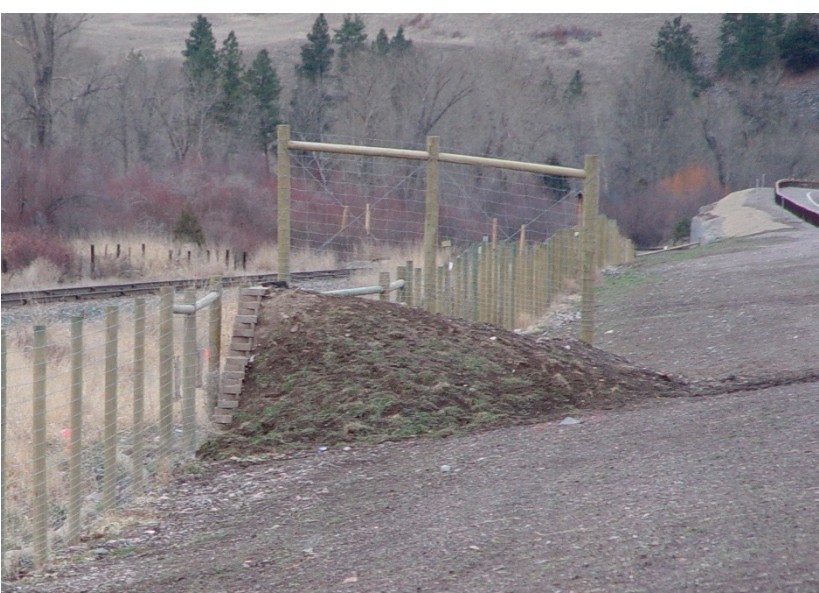

**Figure 10.** Wildlife jump-out and exclusion fencing along US Highway 93, Montana, USA. These structures are used to allow animals to jump down to the safe side of the fence and remove animals from the right-of-way. The wood elements in jump-outs and fencing can be replaced with recycled plastic FRP materials to better withstand the environmental elements and have a longer life cycle. Photo credit: Rob Ament.

Wildlife jump-outs, fencing, and barriers are not required to withstand large structural loads. The fence posts used to line the road are commonly constructed from wood posts and steel, and barrier fences and jump-outs can also be made from concrete, wood and/or stone. All require routine maintenance and replacement. However, these components, which are used in tandem with overpass structures can be replaced with FRP equivalents made of 100% recycled plastics. Replacing these components—fence posts and jump outs—with FRP materials has many benefits, including long-term sustainability and resistance to environmental elements. For example, if fence posts are spaced 4 m apart on both sides of the road, each fence post is 3.5 m in height and 10 cm in width, this results in approximately 13.7 cubic meters of plastic per kilometer (km) that can be removed from landfills and repurposed into exclusionary wildlife fencing to have positive ecological effects on the surrounding ecosystem.

## 4. Existing Guidance for Fiber-Reinforced Polymer Infrastructures in Transportation

AASHTO has developed a guide specification for designing FRP pedestrian bridges [41] and CFFT bridges [42]. These specifications make recommendations for scope of work, dimensions of structural members, design loading, and fabrication. AASHTO is currently working on specifications for composite decks and seismic retrofitting. With the increasing interest in FRP products, federal agencies are investing in additional information regarding application, design, and recommendations within infrastructure [18,30,38,43]. The Federal Highway Administration (FHWA) has also revised its regulations to provide greater flexibility for States to use proprietary or patented materials in Federal-aid highway projects [44]. This ruling allows for more flexibility in selection of materials and encourages innovation in transportation technology and methods.

Most state and provincial transportation agencies have used FRP materials since the mid-nineties. The type of construction project determines whether FRP applications are experimental or a standard practice [45]. The use of FRP wraps to strengthen existing concrete structures and bridge decks made of FRP are the two most accepted practice within transportation agencies. Other uses include pultrusion pedestrian bridges, box culverts, bridge girders, piers, piles, bridge abutments, drains and curbs [18,30]. Each agency is responsible for regulating the use of FRP in infrastructure. Current designs must be signed off by professional engineers and must accommodate all current guidelines for the state/province where the construction projects take place. Some U.S. states, including California, Florida, Kansas, Maine, Michigan, Missouri, Nebraska, Oregon, and Washington, have developed guidelines for the uses of FRP within their jurisdictions.

## 5. Discussion

The technology exists to use FRP materials in applications of highway mitigation measures that reduce WVCs and increase connectivity for wildlife. FRP pultrusion pedestrian bridges and CFFT bridges are approved by AASHTO and are currently used within the U.S. transportation network. The current design elements of wildlife crossings (i.e., geometric dimensions, inclusion of a landscape surface integration with adjacent habitat, sight and sound reduction) all require modifications to accommodate FRP materials in wildlife crossing designs. The application of these resilient materials to wildlife crossings can produce adaptive, economical and effective structures that have the ability to last over twice as long as conventional systems.

The use of pultrusion FRP materials and structures to replace deteriorating steel-concrete-wood pedestrian bridges in Europe is a new strategy that is being widely deployed [46]. There is now an increasing interest in North America because of FRPs high strength-to-weight ratio, durability, and low maintenance cost. Free-formed FRP bridge designs have also become popular around Europe but have yet to be tested to meet national standards in North America. Additional research and testing are needed on the structural properties of these materials as they are applied to designs for wildlife crossings. The current lack of standard design, planning and procurement procedures presents an obstacle for wider adoption of wildlife crossing structures generally, and FRP bridges specifically across North America [13]. The development and execution of strategic guidelines will help fill knowledge gaps and reduce the exposure to professional liability that is associated with insufficient design standards. Sample design calculations and commentary for less common uses and free-formed FRP systems are especially needed. Collaboration between engineers, landscape architects and ecologists are needed to develop and test landscape applications for FRP systems, such as green roof technologies and geotextiles to lighten soil burdens. Similarly, more FRP structures and guidelines are required to make commercial FRP products readily available at reduced costs.

The resins used in FRPs can be altered to ensure the structure has the specific properties needed to resist the environmental factors at its site-specific location. This ensures that FRP wildlife crossing structures will have the longest life-span possible by delaying the degradation of the material's strength. Material testing on glass and carbon FRP shows that after 1000 hours of exposure to environmental conditions (e.g., fresh and saltwater, dry heat, alkali, freeze-thaw cycles, UV, and gasoline fuel) there was less than a 10% change in the elastic properties, and the change in tensile strength was less than 15% when comparing mean values [47]. Different types of UV-absorbing and stabilizing agents can be added to the resin to further protect the FRP material from degradation. Adding zinc and titanium dioxide nanoparticles reduces degradation to only 5% after a week of UV exposure [48]. Furthermore, these tests commonly expose FRP materials to levels of UV exposure not found on earth, i.e., short wavelengths less than 290 nanometers (nm). Longer wavelengths of 365 nm were found to be incapable of inducing a chemical change in high molecular weight polymer structures [49].

Although petroleum-based resins commonly used in FRP materials are arguably unsustainable, research for the use of bio-based resins and fibers in FRP structures are becoming increasingly common. Bio-based polymers are synthetic materials that are processed from vegetable products (e.g., starch,

proteins, and oils). These products are commonly derived from soy beans, potatoes, corn, and flax, but can also be derived from a large number of other grains and seeds [50]. While bio-based resins have a long-life span, they degrade faster than petroleum-based resins. The use of flax, hemp, and jute fibers show comparative mechanical properties to glass but have lower durability [51,52]. Higher moisture, poor compatibility with common resins, and low resistance to fire and ultraviolet (UV) light degradation result in an accelerated deterioration in their mechanical properties. Specific detail is required for each application as each type of bio-based resin are susceptible to different types of biodegradation [17]. Factors other than the origin of natural materials will determine their sustainability and will depend on their application and use. Advancements in bio-based materials life expectancy is required to be competitive with existing conventional petroleum-based FRP methods.

There is strong evidence to suggest that FRP materials are capable of outperforming conventional bridge construction materials in terms of structural performance, maintenance, and cost. Although FRP bridges are becoming more prevalent, they are on average equal, if not more expensive, based on their initial manufacturing and construction costs. Life-cycle costs such as minimal maintenance, durability, and rapid installation tend to be features that justify use of FRP infrastructure [24,28]. However, widespread use has been restricted; this is largely attributed to a lack of knowledge and experience with the use of this material within the civil engineering industry [45,53]. Applications of FRP in wildlife crossing infrastructure provides opportunities for engineers to evaluate this material for low-risk structures.

Further research is required to see how wildlife adapt to these low-profile wildlife crossing structures, especially in cases where a greater diversity of innovative deck surfaces may be desired or used (i.e., green roof technologies) to optimize cost and efficiency of design. It is hypothesized that wildlife will use these structures similarly to current wildlife crossings where landscape and habitat components are similar. Allied professionals such as ecologists, wildlife biologists and landscape architects are important collaborators with engineers in testing, designing, and monitoring wildlife-specific structures.

Through such design and research collaborations, improved and better integrated planning, design, construction and procurement processes can be developed for FRP materials. This in turn will assist in the large-scale deployment of wildlife crossing infrastructure. Furthermore, advocates for wildlife crossings can champion the use of light weight FRP materials, help create demand and increase the scale of production, thus creating reductions in the costs for material transport, abutments, earth-moving, and construction equipment. Because FRP bridges are corrosion resistant and maintain their structural properties over time, owners of the bridge can benefit by reducing costly and time-consuming maintenance. Adapting FRP bridges and structures into effective wildlife crossings can contribute to the long-term goals of improving motorist and passenger safety, wildlife conservation, and cost efficiency, while at the same time reducing plastics from landfills.

## 6. Conclusions

There are many benefits to using FRP materials, compared to conventional methods, for wildlife crossing infrastructure. These composite materials have a high strength-to-weight ratio, allowing for reduced costs in the transportation of materials, construction, and maintenance. The modular approach to current FRP bridge designs applied to wildlife crossing designs, will reduce the amount of time required to manage traffic detours and delays. The light weight FRP materials can also be lifted into place without the use of heavy equipment. Furthermore, the FRP material properties resist environmental deterioration making it adaptable to multiple types of ecosystems and geographic regions. These material characteristics allow engineers to further increase the efficiency of the construction process by reducing the amount of labor required. At the same time, FRP structures will last longer than steel and concrete bridges and require less maintenance.

Although there are limitations to the strength and durability of recycled plastics and bio-based materials for FRP structural components, they can be used for many other design elements for wildlife

crossings. These non-structural applications (e.g., jump-outs, fence posts, sound and light barriers) can be built using bio-based and recycled plastic FRP materials.

Modernizing wildlife crossings to incorporate FRP materials into infrastructure and related design elements will significantly improve current practices.

**Author Contributions:** Conceptualization, R.A., N.-M.L.; methodology, M.B., D.F., R.A., N.-M.L.; investigation, M.B., D.F.; resources, M.B.; writing - original draft preparation, M.B.; writing - review and editing, M.B., D.F., R.A., N.-M.L.; funding acquisition, R.A., N.-M.L. All authors have read and agreed to the published version of the manuscript.

**Funding:** This research was funded in part by the Social Sciences and the Humanities Research Council of Canada (SSHRC) through a federal partnership development grant (grant #890-2015-0029) at Ryerson University in Toronto. Funding was also provided by TPF-5(358), Transportation Pooled Fund Study, administered by the Nevada Department of Transportation and the Small Urban, Rural, and Tribal Center on Mobility (SURTCOM).

**Acknowledgments:** The authors appreciate additional support provided through the ARC partnership and its members including especially the Western Transportation Institute-Montana State University and The Center for Large Landscape Conservation, in Bozeman, MT, and the Woodcock Foundation in New York.

**Conflicts of Interest:** The authors declare no conflict of interest. The funders had no role in the design of the study; in the collection, analyses, or interpretation of data; in the writing of the manuscript, or in the decision to publish the results.

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
