# Peer review of "The Use of Fiber-Reinforced Polymers in Wildlife Crossing Infrastructure"

_sustainability, doi:10.3390/su12041557_

Round 1

Reviewer 1 Report

This review article proposed using fiber reinforced polymers for wildlife crossing infrastructure. FRP should be valuable for this application but its usage is limited due to the lack of knowledge/experience in civil engineering industry. I think this is a well written article and should be valuable for civil engineers as a good introductory material to FRP. I only have some minor comments for the authors to consider.

As the polymer materials tend to creep, is there any long term study of polymer composites for construction/structural applications? Especially under natural conditions (water, temperature, UV, etc) polymer material may start to degrade from surface to internal. Sometimes polymer materials may release additives to the surface due to migration. Will that be an issue to the environment?

Reviewer 2 Report

This review introduces use of FRP in wildlife crossing infrastructure in detail. The subject of this review fits the journal. The manuscript has enough contents for publication as a review. If possible, it is advisable to add further up-to-date information on the use of bio-based and recycled materials. 

line 343: …pour compatibility… ---> poor compatibility ?

Reviewer 3 Report

The paper concerns the benefits and limitations of FRP's use in crossing designs and how they can be applied to many different types of wildlife crossing infrastructure; identifies design-based opportunities and procedures for incorporating recycled plastic material into wildlife crossing infrastructure designs; and evaluates and articulates the political and administrative processes that will help facilitate the adoption of plastic bridge designs by federal, state and local transportation agencies in North America.

In general, the main structure of the paper is correct. In the article there are some shortcomings, which should be corrected before acceptance the paper for publication. Detailed comments are listed below:
Figure 3 is hard to understand and follow to authors idea of the general methods of research in this area. It should be more explanation how pultrusion members are formed and how vacuum assisted resin transfer molded structures are formed. What is exactly on fig. 7 and 9?

Additional comments should be added in regard to the practical value of this research, how the industry can profit from that.

Round 2

Reviewer 3 Report

Authors corrected article. Right now article should be published in the Journal.